# Characteristic-Aroma-Component-Based Evaluation and Classification of Strawberry Varieties by Aroma Type

**DOI:** 10.3390/molecules26206219

**Published:** 2021-10-14

**Authors:** Lixia Sheng, Yinan Ni, Jianwen Wang, Yue Chen, Hongsheng Gao

**Affiliations:** College of Horticulture and Plant Protection, Yangzhou University, Yangzhou 225009, China; wendy_ni0309@163.com (Y.N.); jwwang@yzu.edu.cn (J.W.); qq18852726371@163.com (Y.C.); hsgao@yzu.edu.cn (H.G.)

**Keywords:** *Fragaria × ananassa*, VOCs, characteristic aroma components, aroma type

## Abstract

The unique fruity aroma of strawberries, a popular fruit of high economic value, is closely related to all the volatile organic compounds (VOCs) contained within them. Despite extensive studies on the identification of VOCs in strawberries, systematic studies on fruit-aroma-related VOCs are few, resulting in a lack of effective standards for accurately distinguishing aroma types. In the present study, solid-phase micro extraction and gas chromatography–mass spectrometry were used to analyze and identify VOCs in the ripe fruit of each of the 16 strawberry varieties at home and abroad and to explore their characteristic aroma components and the classification of such varieties by aroma type. The results suggested remarkable variations in the types and contents of VOCs in different strawberry varieties, of which esters were dominant. The principal volatile components, consisting of four esters, three alcohols, one aldehyde, and one ketone, in 16 strawberry varieties were detected based on the absolute and relative contents of VOCs in the fruit. The characteristic aroma components in strawberries, containing nine esters, six aldehydes, and one alcohol, were determined based on the aroma values of different VOCs, and the characteristic aroma components were divided into five types further based on aroma descriptions. Sixteen strawberry varieties were finally divided into four aroma types, namely, peachy, pineapple, fruity, and floral, based on the contributions of different types. The results provided a basis and standard for classifying strawberries by aroma type, studying the hereditary regularity of the fruity aroma of strawberries, and improving aroma quality.

## 1. Introduction

Strawberry (*Fragaria × ananassa* Duch.), a fruit with a pleasant aroma and a high nutritive value [1,2], enjoys great popularity due to its intense fruity aroma [3]. It currently tops the list of the world’s small berry production and is widely distributed, with the highest production in Asia, followed by America and Europe. The World Center of Strawberries is also being gradually shifted from Europe to Asia [4]. There are differences in the cultivation environment, management model, consumer demand, and the quality characteristics of the main strawberry varieties selected in different countries and regions, mostly prominently between those from Japan, European, and American countries. Japanese varieties are highly sweet with a strong aroma but are not suitable for storage or transport, and demonstrate poor disease resistance; European and American varieties are large-fruited and high-yield products, show high disease resistance, and are suitable for storage and transportation but have a high acidity and a flat aroma [5]. With the continuous improvement of people’s requirements for fruity aromas in recent years, new varieties combining the quality characteristics of Japanese, European, and American strawberries, such as “Jingzangxiang” and “Jingtaoxiang,” are being cultivated [6] and have been well received by consumers due to their unique, fruity aroma.

Fruity aromas are derived from a complicated mixture of volatile organic compounds (VOCs). More than 979 VOCs have been identified from fresh strawberries, and 659 species (67%) have been listed only once, that is, they have been reported in only one study [7]. The content of VOCs accounts for only 0.001–0.01% of fresh fruit weight but greatly influences aroma quality [8]. Generally, only a few of these VOCs contribute to fruity aromas, and these unique VOCs are characteristic aroma components [9]. Many VOC compounds have been identified in strawberries but these do not fully clarify the characteristic aroma and aroma qualities of strawberries. The contribution of a VOC to a characteristic aroma rests with the aroma threshold value (ATV) and the concentration of the VOC [10].

The detection of VOCs relies on reliable methods. New technologies have emerged based on electronic sensors and computational identification methods [11]. The E-nose, based on an MOS-type sensor, is a widely used sensor for VOCs detecting and has been found to work well in the quality control and authenticity identification of vegetable oils and essential oils [12,13,14,15,16]. In association with the use of electronic sensors and electronic data, computational signal identification methods such as the use of machine learning methods and the k-NN classification model, have made the data analysis process fast and repeatable [11,15]. However, non-invasive detection and artificial intelligence recognition methods cannot replace the use of a gas chromatograph–mass spectrometer (GC-MS) in precise quantifications. Most quantification studies of complex organic compounds have depended on the high resolution provided by the GC-MS method [13,17]. Most of the current studies on the fruity aroma of strawberries still focus on the determination of VOCs and inter-varietal comparative analysis [18,19,20,21,22,23]. However, the identification and evaluation of characteristic aroma components in strawberries have been less well studied [24], and the classification of strawberries by aroma type is unclear due to the lack of standards for classification. In the present study, 16 strawberry varieties were used as materials to test and analyze their VOCs, principal volatile components (PVCs) were detected and further combined with aroma value (AV) calculations, and the characteristic aroma composition of strawberry varieties was determined, with their aromas divided into different types. The results may offer insights into PVCs in different strawberry varieties. The proposed classification based on characteristic aroma components provides a reference for the evaluation and classification of strawberries by aroma type, and will lay a foundation for improvements in aroma quality and the selection of new varieties.

## 2. Materials and Methods

### 2.1. Materials

Sixteen strawberry varieties listed in Table 1 were planted in the Modern Agriculture Demonstration Park of Guangling District, Yangzhou City, Jiangsu Province, China, in March 2019. All the plants were planted in a soilless cultivation substrate, mixed with vermiculite and perlite. The shed temperature was kept above 25 °C in daytime and above 18 °C in night. The light intensity was kept at 5000 lx with a 16 h/8 h light and dark cycle. All the fruits of the same order of the same plant were picked at harvestable maturity (90% matured, with the same-order fruits of the same plant exhibiting the characteristics of each variety), and those of three clones of each variety were harvested for subsequent determinations.

### 2.2. Determination of VOCs in the Fruit

VOCs were determined on the date the fruits were harvested, as specifically provisioned by Fu Lei et al. [25] with slight changes, via SPME (100 μm Carboxen/PDMS Supelco, St. Louis, MI, USA). After aging for 2 h inside a gas chromatography injector at 250 °C, strawberries were sampled at 10 g each and poured into a 10 mL sample vial (Supelco, St. Louis, MI, USA) after being blended with 3 g NaCl and 5 μL internal standard substance (0.01% 3-nonanone). After the lid was closed properly, SPME was inserted into the headspace part of the sample vial and adsorbed for 30 min in a water bath at 40 °C Then, SPME was inserted into a DSQ gas chromatograph—mass spectrometer (GC-MS, Thermo Corporation, Santa Clara, CA, USA) and observed for 1 min at 240 °C, and all the data collected were analyzed by GC-MS. Each sample was determined three times. Chromatographic conditions: He was used as the carrier gas at a flow rate of 1 mL·min^−^^1^. The heating program was maintained for 2 min at 40 °C, up to 120 °C at a rate of 4 °C·min^−1^, and up to 240 °C at a rate of 10 °C·min^−1^, and then held for 6 min. Mass spectrometry conditions: the GC-MS interface temperature was 240 °C, the ion source temperature was 200 °C, the ionization method was EI, electron energy was 70 eV, and the mass scan range was 33–450 amu.

### 2.3. Identification and Contents of VOCs

According to the total ion chromatogram of VOCs in strawberries, two mass spectral libraries, the NIST library and Wiley library, were retrieved using a computer. Components with a similarity match for forward—reversal retrievals greater than 800 were reserved as candidate identification results, and artificial atlas analysis and literature analysis were further combined to determine the final components.

Absolute content of each VOC (ng/g) = [peak area of each component/peak area of internal standard substance × concentration of internal standard substance (ppm) × 5 μL]/sample amount (g).

Relative content of each VOC = absolute content of a component in a sample (ng/g)/sum of the absolute contents of all components in a sample (ng/g).

Identification of PVCs: in a strawberry variety, the relative content of VOCs [26] was three times higher than that of any other single component. For any other single component, the relative content of PVCs were three times higher than those of VOCs.

### 2.4. Characteristic-Aroma-Component-Based Classification of Strawberries by Aroma Type

AV refers to the ratio of the absolute content of a certain VOC to the ATV of a particular VOC [27]. Gemert indicated that the smaller the threshold concentration of a VOC, the greater the contribution of this VOC to the AV [28].

Note classification: All VOCs in a strawberry variety (with an AV greater than 1) were defined as characteristic AV in the fruit. According to the Aroma Description of Characteristic Aroma Components method [29] and the AV value, VOCs with similar descriptions were merged into the same note.

The AV of a note refers to the sum of the AVs of all the characteristic aroma components that constitute the same note.

For the determination of aroma type, different types were mixed to form the aroma type of a strawberry variety.

For the note-based cluster analysis of strawberry varieties, the AV of each of the types contained in each strawberry variety was calculated first, the data of the AV of each note were converted to the Log2 of the value and were normalized, and cluster analysis was performed using TBtools1.024 (South China Agricultural University, Guangzhou, China).

In the note-contribution-based classification of strawberry varieties by aroma type, the note contribution refers to the sum of the relative proportions of the AVs of the characteristic aroma components contained in a strawberry variety. The relative proportion of the AV of a characteristic aroma component (%) = the AV of each characteristic aroma component/the sum of the AVs of all characteristic aroma components contained in the sample, where the sum of the AVs of all characteristic aroma components contained in the sample refers to the sum of the AVs of all the VOCs in the strawberry variety. A note contribution map was drawn using OriginPro 8.6 (OriginLab, Northampton, MA, USA) and Adobe XD (Oracle, Redwood City, CA, USA). This shows that the higher the proportion of a note, the greater is its contribution to the aroma type. The note with the highest proportion was defined as the aroma type of the corresponding strawberry variety.

## 3. Results and Analysis

### 3.1. VOCs in 16 Strawberry Varieties

Aroma and intensity differed with strawberry varieties, and the types and contents of VOCs in strawberry varieties were substantially different. The absolute contents of VOCs in 16 strawberry varieties are provided in Appendix A, and 67 VOCs, including 35 esters, 7 alcohols, 7 aldehydes, 5 ketones, 7 acids and 6 others were identified (Figure 1). The VOCs in S8 (Jingtaoxiang) were the most diverse, up to 30, whereas those in S9 (Jiuhong) were the least diverse, at merely 18. Among the varieties, esters were the most diverse, accounting for 34–70% of the total VOCs detected. Esters were the most abundant in S5, up to 20. Ketones were the least abundant in strawberries, and neither S9 nor S15 contained ketones, but ketones in the remaining varieties accounted for only 4–9%.

The contents of VOCs differed with strawberry varieties (Figure 2). The total content of VOCs in S7 was the highest, up to 2128.04 ng/g, and that in S14 was the lowest, at only 132.49 ng/g. The proportions of esters were the highest among all VOCs contained in the varieties S5, S7, S14, S13, S3, S8, and S47, at 91.43–98.94%; alcohols in the varieties S10, S16, S15, S9, S11, and S16 had relatively large proportions. Aldehyde and ketone levels were low in most strawberry varieties but were high in a few strawberry varieties. For example, the total contents of aldehydes were 288.15 and 182.95 ng/g in S2 and S12, respectively, and the total content of ketones in S6 was 133.6 ng/g.

### 3.2. PVC Spectra and Characteristic Aromas of 16 Strawberry Varieties

Based on the absolute and relative contents of VOCs in 16 strawberry varieties (Appendix A), the PVC spectra of 16 strawberry varieties, containing four esters, three alcohols, one aldehyde, and one ketone were selectively established (Table 2). The relative contents of nerolidol accounted for 23.69–37.56% of alcohols in the varieties S3, S10, S11, and S16. Linalool was one of the PVCs in S15 (relative content: 23.47%). γ-decalactone was an ester and a PVC in S8, S13, and S14, with its relative content ranging from 33.69% to 49.23%. Trans-2-hexenal was a PVC in S2, S4, and S12, with relative contents of 40.79% (S2), 30.02% (S4), and 32.25% (S12), respectively. Several VOCs, such as methyl hexanoate, octyl acetate, and 1-octyl hexanoate, showed higher relative contents in individual varieties.

AV fully considers the contents and ATVs of VOCs, and can well characterize an aroma, approaching an expression of its taste in the human body. When the AV is greater than 1, the corresponding VOC constitutes a characteristic aroma component in the fruit. Among the 68 VOCs detected, 25 VOCs of fatty acids had no ATV. Based on the ATVs, concentrations, and aroma descriptions of 43 VOCs (Appendix A), 16 characteristic aroma components were detected in 16 strawberry varieties (Appendix A), containing nine esters, six aldehydes, and one alcohol. The contributions of several VOCs in different varieties were quite diverse, and even the characteristic aroma components within the same varieties were different. For example, the AVs of γ-decalactone, a characteristic aroma component, were 235.90 and 1.20 in two different varieties, S13 and S12. Eight characteristic aroma components were detected in S15, and the AVs of linalool and nonanal were 1116.8 and 2.17, respectively. Linalool was a common characteristic aroma in the 15 remaining strawberry varieties in the present study (except for S7), with the AV ranging from 23.8 (S1) to 1116.8 (S15), having considerable differences. Moreover, 12 characteristic aroma components were detected in S12, whereas only three ones, namely, methyl butyrate (1.14), γ-decalactone (93.19), and linalool (30.88), were found in S14. Decanal (1.48) is a unique characteristic aroma component detected only in S12, in which characteristic aromas were the most diverse.

### 3.3. Classification of 16 Strawberry Varieties by Aroma Type

According to the aroma descriptions of volatile compounds (Appendix A), characteristic aroma components in 16 strawberry varieties were classified into five types, namely, pineapple-like, fruity, citrus-like, peachy, and floral (Appendix A). Ethyl butyrate, methyl hexanoate, and ethyl hexanoate were found in pineapple-like notes; γ-undecalactone, γ-decalactone, and octyl acetate were detected in peachy notes; methyl butyrate, hexyl acetate, ethyl isobutyrate, and hexanal were contained in fruity notes; linalool, trans-2-hexenal, and lauric aldehyde were included in floral notes; and n-nonanal, decanal, and methyl anthranilate were part of citrus-like notes.

The cluster analysis on 16 strawberry varieties was performed based on the AVs of the five types, obtaining two different, highly supported groups (Figure 3). First, S5, S13, S8, S14, and S7 were all separated from the 11 other strawberry varieties because they contained peachy compounds with high AVs and floral compounds with low AVs. Next, S7 was separated from Group B1 because its pineapple-like compounds with high AVs were different from Group B2 (S5, S13, S8, and S14). Differences were also observed among 11 varieties in Group A in aroma type, and these varieties were divided into two subgroups. S1 and S6 were classified into Group A1 because they were rich in fruity compounds with high AVs and floral compounds with low AVs, suggesting a closer relationship between them in aroma type. Nine strawberry varieties of Group A2, having differences in the AVs of pineapple-like and floral compounds, were classified into three groups: S2, with a lack of pineapple-like compounds, was classified into Group A21; floral compounds in Group A22 (S4, S9, S15, and S16) were higher than those in Group A23 (S3, S10, S11, and S12), causing them to be separated from one another.

Different types have different contributions to the aroma types of strawberries because each note contains multiple VOCs, and each compound plays a different role in the aroma of strawberries. Figure 3 shows that the contribution of citrus-like notes in the fruity aroma of strawberries is evidently less than that of the other types and does not play any decisive role in the fruity aroma type. Therefore, strawberry fruity aromas were further classified into four types based on the corresponding aroma contributions, and 16 varieties were divided into four groups (Figure 4).

Group 1: Peachy

Group 1 provides an outstanding fruity peach-like aroma. γ-decalactone has a pleasant fruity aroma like that of a peach. Relatively high levels of γ-decalactone were detected in S8, S13, S14, and S5. However, because the threshold concentration is relatively low, it also plays an important role in sending out a peach-like aroma, even though its content is low. Moreover, high levels of octyl acetate were detected in S5. Octyl acetate has a slightly bitter fruity aroma, which reminds one of peach.

Group 2: Pineapple

Group 2 outstandingly provides a fruity aroma of pineapple. The aroma description shows that the three VOCs in strawberries—ethyl butyrate, methyl hexanoate, and ethyl hexanoate—all have a pineapple-like aroma. S7 contains higher levels of high-content ethyl butyrate, methyl hexanoate, and ethyl hexanoate, with AVs substantially higher than those of the other varieties and higher proportions of the corresponding aroma types.

Group3: Fruity

Group 3 maintains fruity aromas of apple, apricot, or pear. These fruity aromas are constituents of the basic aroma of strawberries, and the corresponding AVs are more apparent in all varieties. Methyl butyrate has an apple aroma and a corresponding sweetness; hexyl acetate has a pleasant fruity aroma, which is suggestive of pears; and hexanal contains a special fruity aroma. The three compounds were all found in 16 varieties, and they are the principal characteristic aroma components of strawberries. Moreover, ethyl isobutyrate, with a fruity aroma, is a characteristic aroma component in S1 and S6, and it is the main contributor to the classification by fruity aroma type.

Group 4: Floral

Group 4 contains floral aromas such as rose and jasmine. It contains nine varieties, S2, S3, S4, S9, S10, S11, S12, S15, and S16. Linalool, with a high AV, was detected in all the strawberry varieties. Linalool has a typical pleasant aroma of flowers and is a PVC of the floral note of strawberries. Methyl anthranilate has a special aroma of orange flower and is a contributor to the classification by floral aroma type.

## 4. Discussion

Strawberry is considered the fruit with the most diverse fruity aroma components. Over the past decades, more than 979 VOCs have been identified in strawberries [7]. In the present study, VOCs in 16 strawberry varieties were detected and analyzed, and 68 VOCs were identified. The results showed that esters are the most abundant compounds in all the strawberry varieties. The total content of esters is also the highest, followed by alcohols and aldehydes, and the total content of ketones is the lowest. The findings are consistent with a previous report [30]. Interestingly, *F. vesca*, an ancestor of *F. x ananassa,* is also richer in esters, and there were no significant differences in the average relative content of esters between wild strawberries in other reports and the cultivated strawberries in this study [31]. Moreover, the typical ‘wild-strawberry-like’ aroma monoterpenes, such as α-pinene, β-myrcene, α-terpineol, and α-phellandrene, that are common in wild strawberries were not listed among the VOC profiles of cultivated strawberries [32,33]. Different VOCs give fruits varying sensory properties. Only some of the volatiles in fresh-eating strawberries play a major role in the fruity aroma and the overall aroma of the strawberry. In the present study, eight VOCs, namely, methyl butyrate, methyl caproate, heptyl acetate, γ-dodecanolactone, hexanal, E-2-hexenal, 2,5-dimethyl-4-methoxyl-3(2H)-furanone (DMMF), and hexanoate, were identified in 16 strawberry varieties, and high levels of ethyl hexanoate, methyl caprylate, linalool, and nerolidol were detected in 12 varieties. Eleven VOCs, 1-octen-3-yl acetate, n-octyl formate, methyl benzoate, ethyl benzoate, decyl acetate, lavendulyl acetate, octyl octanoate, malonic acid, heptadecane, 1-methylethyl benzoate, and cis-3-hexenyl acetate, were listed as the only components present in certain strawberry varieties. These unique components may contribute to the distinctive aromas of different strawberry varieties.

The fruity aroma of a strawberry variety is a combination of aromas such as caramel, jam, floral fragrance, fruity aroma, buttery, sour, and herbal flavor [34,35,36]. DMMF and 2,5-dimethyl-4-hydroxyl-3(2H)-furanone (DMHF) smell like “caramel” [37], and their AVs [1] are very high. The contents of such volatiles are also high in certain strawberry varieties [2]. However, DMMF (instead of DMHF) was detected in several studies [24,38]. This finding is consistent with the results of the present study. This finding may be caused by the concentrations of furans fluctuating markedly during the entire ripening process of the strawberry due to metabolism [30] or due to the extraction of volatiles from strawberries by means of the boiling water bath heating method, resulting in the loss of several compounds [10].

In the present study, strawberry varieties were classified into four aroma types by their characteristic aroma: peachy, pineapple, fruity, or floral. In terms of the peachy type, domestic and foreign studies have shown that a series of lactones, particularly γ-decalactone, are the main contributors to this [39,40]. The relative content of γ-decalactone in S8 is highly substantial, providing a pleasant peach aroma [24]. S5 also has a strong fruity aroma of peach. The present study showed that the content of γ-decalactone in S5 is lower than that in S8, S13, and S14, but its content of octyl acetate is very high. This volatile has a fruity aroma of peach and is a unique component in S5 compared with the three other peachy types. Methyl caproate is reported as one of the main ingredients of the pineapple type [41], and methyl butyrate and ethyl hexanoate have been described as having a pineapple aroma [29]. In the present study, the total content and AV of components of the pineapple type in S7 provide important references for classifying the variety into aroma types.

The classification of a strawberry variety by aroma type may differ in studies. Wang Juan et al. classified S1 into the floral type and S4 into the pineapple type [24]. However, in the present study, S1 and S4 fell into the fruity aroma type and the floral type, respectively. This may be because the former findings were obtained through sensory tasting, whereas the latter ones were obtained by assessing the AV and note contributions of each of the VOCs in the fruit. In the present study, the AVs of the VOCs of the floral type in S1 were second only to those of the fruity aroma type, and the VOCs of the pineapple type in S4 contributed together with those of the floral type, forming the unique aroma. Surely, such factors as growing environment conditions, cultivation measures, and maturity may result in several differences in the aroma performance of the same strawberry variety. Overall, the classification of strawberries based on the characteristic aroma components in the present study is more rigorous and repeatable than subjective judgments or sensory tasting, providing an important reference for subsequent studies on the genetic laws of fruity aroma, and the selection and aroma quality improvements of strawberry varieties.

## Figures and Tables

**Figure 1 molecules-26-06219-f001:**
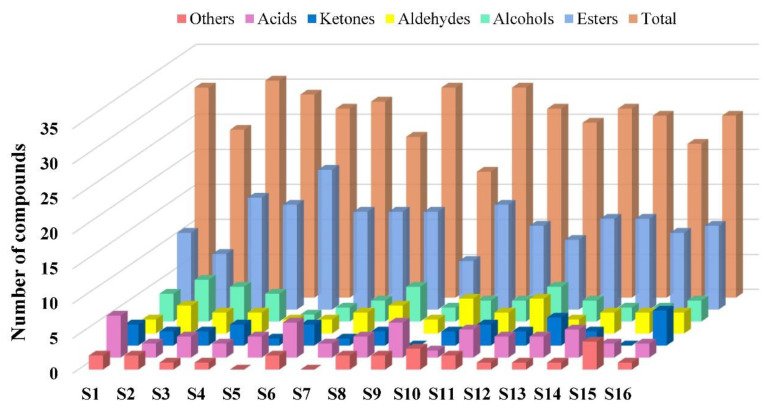
The number of compounds in 16 strawberry varieties. S1 (Benihoppe), S2 (Ssanta), S3 (Akihime), S4 (Snow White), S5 (Tokun), S6 (Yuexin), S7 (Jingzangxiang), S8 (Jingtaoxiang), S9 (Jiuhong), S10 (Seolhyang), S11 (Sachinoka), S12 (Hongyu), S13 (Yanli), S14 (Zaoyu), S15 (Rongmei3), S16 (106).

**Figure 2 molecules-26-06219-f002:**
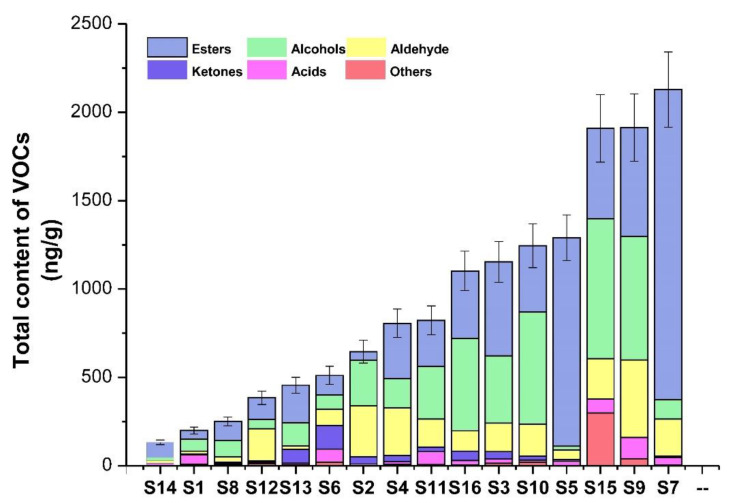
Total content of VOCs in 16 strawberry varieties. S1 (Benihoppe), S2 (Ssanta), S3 (Akihime), S4 (Snow White), S5 (Tokun), S6 (Yuexin), S7 (Jingzangxiang), S8 (Jingtaoxiang), S9 (Jiuhong), S10 (Seolhyang), S11 (Sachinoka), S12 (Hongyu), S13 (Yanli), S14 (Zaoyu), S15 (Rongmei3), S16 (106). The error bars indicate the standard deviation values.

**Figure 3 molecules-26-06219-f003:**
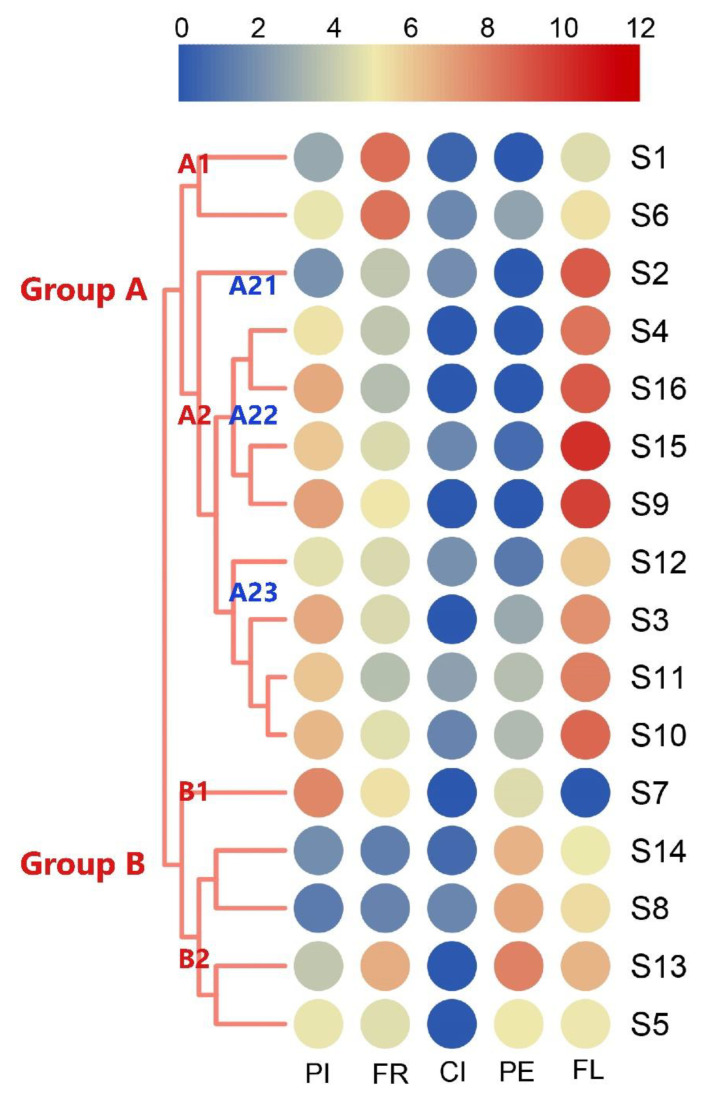
Cluster analysis of 16 strawberry varieties based on aroma type. PI: pineapple-like, FR: fruity, CI: citrus-like, PE: peachy, FL: floral. The color blocks represent the normalization aroma value of the scent converted by Log2. S1 (Benihoppe), S2 (Ssanta), S3 (Akihime), S4 (Snow White), S5 (Tokun), S6 (Yuexin), S7 (Jingzangxiang), S8 (Jingtaoxiang), S9 (Jiuhong), S10 (Seolhyang), S11 (Sachinoka), S12 (Hongyu), S13 (Yanli), S14 (Zaoyu), S15 (Rongmei3), S16 (106).

**Figure 4 molecules-26-06219-f004:**
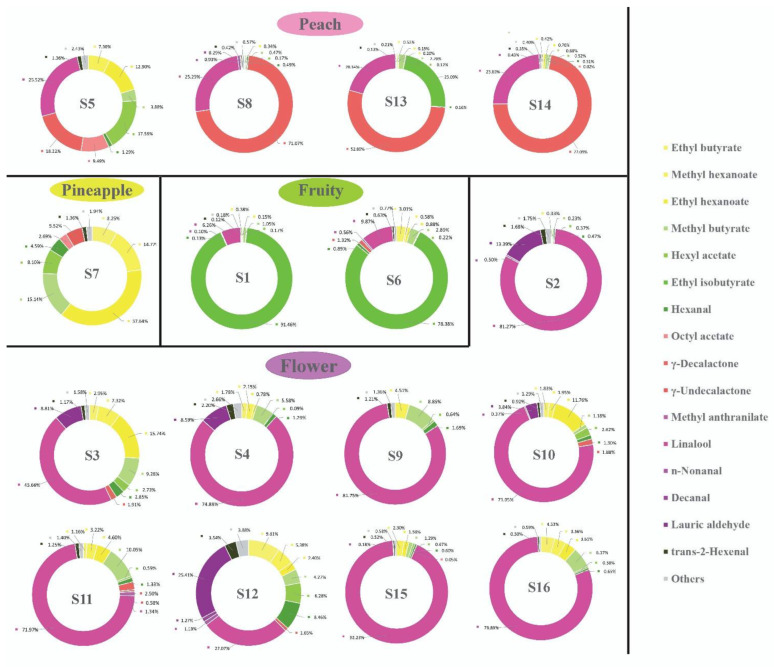
Note-contribution-based classification of 16 strawberry varieties by aroma type. The circle graph represents the sum of the relative proportions of all AVs (100%), and the numbers represent the relative proportions of all AVs of all VOCs. Different characteristic aroma components of a same note are distinguished by different shades of a same color, and noncharacteristic aroma components are incorporated into ‘Others’. S1 (Benihoppe), S2 (Ssanta), S3 (Akihime), S4 (Snow White), S5 (Tokun), S6 (Yuexin), S7 (Jingzangxiang), S8 (Jingtaoxiang), S9 (Jiuhong), S10 (Seolhyang), S11 (Sachinoka), S12 (Hongyu), S13 (Yanli), S14 (Zaoyu), S15 (Rongmei3), S16 (106).

**Table 1 molecules-26-06219-t001:** Information on 16 strawberry varieties.

Number	Varieties of Samplings	Source
S1	*Fragaria × ananassa* ‘Benihoppe’	NARO Institute of Vegetable and Tea Science (NIVTS), Tsukuba, Japan
S2	*Fragaria × ananassa* ‘Ssanta’	Beijing Academy of Agriculture and Forestry Sciences (BAAFS), Beijing, China
S3	*Fragaria × ananassa* ‘Akihime’	NIVTS, Tsukuba, Japan
S4	*Fragaria × ananassa* ‘Snow White’	BAAFS, Beijing, China
S5	*Fragaria × ananassa* ‘Tokun’	NIVTS, Tsukuba, Japan
S6	*Fragaria × ananassa* ‘Yuexin’	Zhejiang Academy of Agricultural Sciences, Zhenjiang, China
S7	*Fragaria × ananassa* ‘Jingzangxiang’	BAAFS, Beijing, China
S8	*Fragaria × ananassa* ‘Jingtaoxiang’	BAAFS, Beijing, China
S9	*Fragaria × ananassa* ‘Jiuhong’	Jiangsu Academy of Agricultural Sciences, Nanjing, China
S10	*Fragaria × ananassa* ‘Seolhyang’	The Korean Strange Plant Research Institute, Daejeon, Korea
S11	*Fragaria × ananassa* ‘Sachinoka’	NIVTS, Tsukuba, Japan
S12	*Fragaria × ananassa* ‘Hongyu’	Hangzhou Academy of Agricultural Sciences, Hangzhou, China
S13	*Fragaria × ananassa* ‘Yanli’	Shenyang Agricultural University, Shenyang, China
S14	*Fragaria × ananassa* ‘Zaoyu’	Jiangsu Academy of Agricultural Sciences, Nanjing, China
S15	*Fragaria × ananassa* ‘Rongmei3′	Zhenjiang Institute of Hilly Region Agricultural Sciences, Zhenjiang, China
S16	*Fragaria × ananassa* ‘106′	Zhenjiang Institute of Hilly Region Agricultural Sciences, Zhenjiang, China

**Table 2 molecules-26-06219-t002:** The PVCs of 16 strawberry varieties.

Varieties	Compounds Relative Content (%)							
	Methyl Hexanoate	Octyl Acetate	Octyl Hexanoate	γ-Decalactone	Linalool	Nerolidol	Diethylene Glycol	Trans-2-Hexenal
S1	7.04 ± 1.22	-	-	-	4.69 ± 1.11	2.54 ± 0.34	23.98 ± 2.04	6.74 ± 1.07
S2	2.68 ± 0.14	-	-	-	26.32 ± 2.01	11.73 ± 1.21	0.49 ± 0.07	40.79 ± 3.52
S3	20.24 ± 2.74	-	0.84 ± 0.15	0.37 ± 0.01	5.05 ± 0.91	26.38 ± 2.10	0.39 ± 0.04	9.69 ± 1.30
S4	9.76 ± 1.31	-	-	-	13.61 ± 1.32	6.01 ± 0.79	-	30.02 ± 3.03
S5	6.87 ± 1.03	41.99 ± 3.67	13.63 ± 1.81	1.2 ± 0.13	0.96 ± 0.36	-	-	3.85 ± 0.94
S6	4.5 ± 0.56	-	-	0.71 ± 0.06	3.05 ± 1.20	12.36 ± 1.72	-	14.5 ± 1.82
S7	19.57 ± 3.01	16.75 ± 2.98	20.91 ± 2.24	-	-	-	2.05 ± 0.38	5.4 ± 0.88
S8	2.28 ± 0.21	-	-	33.63 ± 2.98	6.82 ± 1.62	29.73 ± 3.12	0.22 ± 0.01	8.5 ± 1.24
S9	23.2 ± 2.98	-	-	-	16.81 ± 2.01	19.69 ± 1.37	-	18.61 ± 2.03
S10	8.04 ± 1.14	-	0.14 ± 0.01	0.54 ± 0.08	11.69 ± 1.37	37.56 ± 3.75	0.89 ± 0.12	11.36 ± 1.50
S11	13.62 ± 1.67	-	0.48 ± 0.03	0.74 ± 0.04	12.16 ± 0.93	23.69 ± 2.42	-	15.81 ± 1.81
S12	16.32 ± 2.16	-	-	0.22 ± 0.01	3.28 ± 0.24	6.87 ± 1.05	-	32.25 ± 3.56
S13	1.9 ± 0.05	-	-	36.23 ± 3.25	7.96 ± 0.88	20.05 ± 2.01	-	3.87 ± 0.45
S14	4.11 ± 0.78	0.61 ± 0.02	-	49.23 ± 4.01	9.32 ± 1.13	3.28 ± 0.56	-	10.08 ± 1.40
S15	14.59 ± 1.43	1.47 ± 0.09	0.87 ± 0.03	-	23.4 ± 2.18	18.05 ± 1.68	-	9.93 ± 1.05
S16	20.42 ± 2.31	-	-	-	18.33 ± 2.04	29.03 ± 2.96	-	8.56 ± 0.87

-: Not found or not present. S1 (Benihoppe), S2 (Ssanta), S3 (Akihime), S4 (Snow White), S5 (Tokun), S6 (Yuexin), S7 (Jingzangxiang), S8 (Jingtaoxiang), S9 (Jiuhong), S10 (Seolhyang), S11 (Sachinoka), S12 (Hongyu), S13 (Yanli), S14 (Zaoyu), S15 (Rongmei3), S16 (106). The relative content is indicated by mean ± SD.

## Data Availability

Not applicable.

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
