# Peer review of "Characteristic-Aroma-Component-Based Evaluation and Classification of Strawberry Varieties by Aroma Type"

_molecules, 2021, doi:10.3390/molecules26206219_

Round 1
Reviewer 1 Report
The reviewed manuscript presented the profile of volatile organic compounds determined in 16 different strawberry varieties. The work presents very interesting research. Moreover, manuscript is very well prepared. Below are some comments.
Additional minor comments:
- The authors should perform a statistical analysis of the results, mainly those shown in Fig 2 and Table 3.
- The results in Table 3 should be presented as mean ± SD.
Author Response
- The authors should perform a statistical analysis of the results, mainly those shown in Fig 2 and Table 3.
- The results in Table 3 should be presented as mean ± SD.
- Answer:We have supplemented statistical analysis (mean ± SD) in Fig 2 and Table 3.
Reviewer 2 Report
Characteristic-Aroma-Component-Based Evaluation and Classification of Strawberry Varieties by Aroma Type.
In the present study, 16 strawberry varieties were used as materials to test and analyze their volatile organic compounds, principal volatile components were detected and further combined with aroma values calculations, and the characteristic aroma composition of strawberry varieties was determined with their aromas divided into different types.I commented on the manuscript and the comments are presented below:
Part 1: Introduction.
The Introduction to the study provides with some general information on research performed by other researchers. The authors should supplement this chapter with newer works, but also with works that deal generally with research obtained from various methods and plants. Biological research material has a huge impact on the obtained results. In turn, such material is influenced by many factors, including soil type, fertilization, cultivation method, environmental parameters (temperature, rainfall, sunlight, air humidity), etc. The supplemented chapter with additional information will certainly bring you closer and make the research goals clearer. The above information can be obtained, for example, in the following works:
„A machine learning method for classification and identification of potato cultivars based on the reaction of mos type sensor-array”;
“Contribution of the ratio of tocopherol homologs to the oxidative stability of commercial vegetable oils”;
“Classification and Identification of Essential Oils from Herbs and Fruits Based on a MOS Electronic-Nose Technology”;
“Development of Perilla seed oil and extra virgin olive oil blends for nutritional, oxidative stability and consumer acceptance improvements”;
“E-nose, e-tongue and e-eye for edible olive oil characterization and shelf life assessment: A powerful data fusion approach”;
„Coupling MOS sensors and gas chromatography to interpret the sensor responses to complex food aroma: Application to virgin olive oil”.
Authors should explain the abbreviation (AV) in the text Introduction (line 61). Although it is used in the abstract, it should be explained in the main text. This is just done on line 108.
Part 2: Material and Methods
The Materials and methods section provides the reader with enough information to repeat the experiments conducted. Biological research material has a huge impact on the obtained results. In turn, such material is influenced by many factors, including soil type, fertilization, cultivation method, environmental parameters (temperature, rainfall, sunlight, air humidity), etc. Then it would be possible to associate the obtained result with a specific chemical compound and on this basis to obtain interesting conclusions. Since the region, temperature, rainfall, and sunlight have a great influence on the composition of the olive, even in the same region, different compositions of the obtained olives are obtained every year.
Part: 3 Results and Analysis
The most part the Results and Analysis section is well structured and the obtained data were subjected for statistical analysis. The results were not fully discussed. A full discussion of the results obtained with other work in this field should be carried out in more aspects. the Results section is well structured and the obtained data were subjected for statistical analysis.
Part: 4 Discussion
The results were not fully discussed. A full discussion of the results obtained with other work in this field should be carried out in more aspects. The authors should supplement the discussion based on publications issued in recent years. Here I have a question about the large amount of alcohol obtained for some varieties. Is it not the case that the fruits of these varieties were already in the initial stage of decomposition and therefore there was a high content of this compound, which is related to the fermentation process? Look at work, e.g. „Detection and measurement of aroma compounds with the electronic nose and a novel method for MOS sensor signal analysis during the wheat bread making process”;
„Potential use of electronic noses, electronic tongues and biosensors as multisensor systems for spoilage examination in foods”. Conclusions are synthetically described and result from the conducted research but the obtained results were not fully discussed in the manuscript.
Part: References.
The literature used is appropriate however the authors should supplement this chapter with newer works.
Author Response
In the present study, 16 strawberry varieties were used as materials to test and analyze their volatile organic compounds, principal volatile components were detected and further combined with aroma values calculations, and the characteristic aroma composition of strawberry varieties was determined with their aromas divided into different types. I commented on the manuscript and the comments are presented below:
Part 1: Introduction.
The Introduction to the study provides with some general information on research performed by other researchers. The authors should supplement this chapter with newer works, but also with works that deal generally with research obtained from various methods and plants. Biological research material has a huge impact on the obtained results. In turn, such material is influenced by many factors, including soil type, fertilization, cultivation method, environmental parameters (temperature, rainfall, sunlight, air humidity), etc. The supplemented chapter with additional information will certainly bring you closer and make the research goals clearer. The above information can be obtained, for example, in the following works:
„A machine learning method for classification and identification of potato cultivars based on the reaction of mos type sensor-array”;
“Contribution of the ratio of tocopherol homologs to the oxidative stability of commercial vegetable oils”;
“Classification and Identification of Essential Oils from Herbs and Fruits Based on a MOS Electronic-Nose Technology”;
“Development of Perilla seed oil and extra virgin olive oil blends for nutritional, oxidative stability and consumer acceptance improvements”;
“E-nose, e-tongue and e-eye for edible olive oil characterization and shelf life assessment: A powerful data fusion approach”;
„Coupling MOS sensors and gas chromatography to interpret the sensor responses to complex food aroma: Application to virgin olive oil”.
Authors should explain the abbreviation (AV) in the text Introduction (line 61). Although it is used in the abstract, it should be explained in the main text. This is just done on line 108.
Answer:
- This study paid attention to difference in the VOCs of strawberry varieties and aimed to be a basic of our further study of genetic regulation on the aroma quality (NOT the evaluation for foods or products). So we didn’t compared the strawberry VOCs in different environmental parameters (temperature, rainfall, sunlight, air humidity). We designed the consistent cultivation method to make sure that VOCs came from the genetic factors not the environmental factors.
- The different detecting technology of organic compounds was not the main topic of our study. Anyway, we have supplemented the different classification methods in the introduction.
- We have explained it (aroma value) in the main text.
Part 2: Material and Methods
The Materials and methods section provides the reader with enough information to repeat the experiments conducted. Biological research material has a huge impact on the obtained results. In turn, such material is influenced by many factors, including soil type, fertilization, cultivation method, environmental parameters (temperature, rainfall, sunlight, air humidity), etc. Then it would be possible to associate the obtained result with a specific chemical compound and on this basis to obtain interesting conclusions. Since the region, temperature, rainfall, and sunlight have a great influence on the composition of the olive, even in the same region, different compositions of the obtained olives are obtained every year.
Answer:
- The environmental influence on VOCs of strawberry were not within the scope of our study. To avoid differential environment as far as possible,the same cultivation medium, cultivation method and conditions were used in the same greenhouse.
- The geographical influence on strawberry VOCs belongs to ecological study and it is closely related with ecotypes. Within the bounds of our knowledge, different ecologicalniche distributions of wild strawberry contribute to different diploid strawberry ecotypes, there was no ecotype differentiation of the octaploid cultivated strawberry due to the intensive cultivation method. Our study only used the octaploid cultivated strawberries.
- We have not explored different compositions of different year in this study, whereas the interannualvariation of VOCs would be the interesting topic of our further study.
Part: 3 Results and Analysis
The most part the Results and Analysis section is well structured and the obtained data were subjected for statistical analysis. The results were not fully discussed. A full discussion of the results obtained with other work in this field should be carried out in more aspects. the Results section is well structured and the obtained data were subjected for statistical analysis.
Answer:We have revised the discussion with more compare analysis of similar study.
Part: 4 Discussion
The results were not fully discussed. A full discussion of the results obtained with other work in this field should be carried out in more aspects. The authors should supplement the discussion based on publications issued in recent years. Here I have a question about the large amount of alcohol obtained for some varieties. Is it not the case that the fruits of these varieties were already in the initial stage of decomposition and therefore there was a high content of this compound, which is related to the fermentation process? Look at work, e.g. „Detection and measurement of aroma compounds with the electronic nose and a novel method for MOS sensor signal analysis during the wheat bread making process”;
„Potential use of electronic noses, electronic tongues and biosensors as multisensor systems for spoilage examination in foods”. Conclusions are synthetically described and result from the conducted research but the obtained results were not fully discussed in the manuscript.
Answer:We have revised the discussion with more compare analysis.
Part: References.
The literature used is appropriate however the authors should supplement this chapter with newer works.
Answer:We have updated the references.